# Role of Low-Risk HPV PCR Monoinfection in Screening for HSIL and Anal Cancer in Men Who Have Sex with Men Living with HIV

**DOI:** 10.3390/ijms24065642

**Published:** 2023-03-15

**Authors:** Carmen María García-Martínez, Inmaculada Calle-Gómez, Javier López-Hidalgo, Patricia Gómez-Ronquillo, Mohamed Omar-Mohamed Balgahata, Carmen Hidalgo-Tenorio

**Affiliations:** 1Service of Internal Medicine, Hospital Universitario Virgen de las Nieves, Instituto de Investigación Biosanitario de Granada (IBS-Granada), 18012 Granada, Spain; 2Service of Pathology, Hospital Universitario Virgen de las Nieves, Instituto de Investigación Biosanitario de Granada (IBS-Granada), 18012 Granada, Spain; 3Unit of Infectious Diseases, Complejo Hospitalario de Jaén, 23007 Jaen, Spain; 4Unit of Infectious Diseases, Hospital Universitario Virgen de las Nieves, Instituto de Investigación Biosanitario de Granada (IBS-Granada), 18012 Granada, Spain

**Keywords:** human immunodeficiency virus (HIV), men who have sex with men living with HIV (MSM-LHIV), human papillomavirus (HPV), non-AIDS-defining diseases, anal canal

## Abstract

To determine the value of low-risk human papillomavirus (HPV) PCR to screen for “high-grade anal squamous intraepithelial lesion and anal cancer” (HSIL-plus), rate of patients with low-grade anal squamous intraepithelial lesion (LSIL) progressing to HSIL-plus, and progression-related factors. Prospective, longitudinal study of consecutive MSM-LHIV attended between May 2010 and December 2021 and followed for 43 months (IQR: 12–76). HIV-related variables were gathered at baseline, performing anal cytology for HPV detection/genotyping, thin-layer cytological study, and high-resolution anoscopy (HRA). Follow-up was annual when HRA was normal or LSIL, and post-treatment in cases of HSIL-plus, re-evaluating sexual behavior, viral-immunological status, and HPV infection of anal mucosa. The 493 participants had mean age of 36 years: CD4 nadir < 200 cells/uL in 23.1%, virological failure in 4.1%, and tetravalent HPV vaccine > 5 years earlier in 15%. HSIL-plus was ruled out in patients with monoinfection by low-risk HPV genotype and normal cytology (100% sensitivity, 91.9% specificity, PPV 2.9%, and NPV 100%). Progression from LISL to HSIL-plus occurred in 4.27% of patients within 12 months (IQR: 12–12): risk factors were acquisition of high-risk (HR: 4.15; 95% CI: 1.14–15.03) and low-risk (HR: 3.68 95% CI: 1.04–12.94) HPV genotypes, specifically genotype 6 (HR: 4.47, 95% CI: 1.34–14.91), and history of AIDS (HR: 5.81 95% CI: 1.78–18.92). Monoinfection by LR-HPV genotypes in patients with normal cytology is not associated with anal cancer or precursor lesions. Progression from LSIL to HSIL-plus, observed in <5% of patients, was related to acquisition of HR and LR HPV genotypes, especially 6, and a history of AIDS.

## 1. Introduction

Anogenital infection by human papillomavirus (HPV) is the most common sexually transmitted infection (STI) worldwide [1]. It has a predilection for skin and mucosal cells and is associated with a wide range of lesions, from warts to low- and high-grade squamous intraepithelial lesions (LSIL and HSIL) and vulvar, vaginal, penis, oropharyngeal, and anal carcinomas [2].

Immunosuppression is a risk factor for HPV acquisition and is produced by infection with the human immunodeficiency virus (HIV). The degree of immunodepression has been related to the incidence of anal dysplasia from chronic infection by HPV genotypes [3,4]. This infection commonly involves the anal mucosa, and anal cancer is one of the most frequent non-AIDS-defining diseases in people living with HIV (PLHIV), mainly in women and men who have sex with men (MSM) [5,6].

Research to date has established a relationship between high-risk (HR) genotypes and a greater likelihood of precursor lesions of anogenital or oropharyngeal cancer/carcinomas [7,8]. MSM living with HIV (MSM-LHIV) who have normal anal cytology and are not infected by an oncogenic HPV genotype have shown no increased risk of HSIL or anal cancer [9]. 

Inadequate data are available on the presence/absence of HSIL or anal cancer in patients with monoinfection by LR HPV genotype. The objectives of this study were to verify the sensitivity (S), specificity (Sp), and negative (NPV) and positive (PPV) predictive value of the presence of anal mucosal infection by LR HPV genotypes to screen for HSIL and/or anal cancer (HSIL-plus) in MSM-LHIV; to determine the percentage of these patients progressing from LSIL to HSIL-plus; and to analyze the factors related to this progression. 

## 2. Results

### 2.1. Baseline Characteristics of the Cohort

The study included 493 MSM-LHIV aged 36 years (±10.36), 48.6% were university students, 48.1% smoked a mean of 1.65 packs/year (IQR: 0–15), 15% had received the full course of tetravalent HPV vaccine; and 39.8% had another STI; the median number of partners to date was 50 sexual partners (IQR: 20–200), and the median number in the previous 12 months was 2 (IQR 1–8). Table 1 exhibits results for the remaining characteristics. 

The mean age at HIV diagnosis was 31 years, and the mean CD4 count was 444.6 cells/uL, with 23.1% of participants having CD4 nadir < 200 cells/uL. At study enrolment, the CD4 count was 689.8 cells/uL and CD4/CD8 ratio 0.7 (0.46–0.98), 10.8% were naïve, and 4.1% were in virological failure. They had received antiretroviral treatment (ART) for a median of 20 months (IQR: 7–73): 32.9% with non-nucleoside analog reverse transcriptase inhibitor (NNRTI), 26.2% with protease inhibitor (PI), and 38.5% with integrase strand transfer inhibitors (INSTI). Table 2 exhibits the findings for other study variables. 

Infection by oncogenic genotypes was detected in 74%, with a median of 1 genotype (IQR:1–3); infection by non-oncogenic genotypes in 71.2%, with a median of 1 genotype (IQR:0–2); and coinfection in 56%. HR genotypes 16 (26.4%) and 31 (13.6%) were the most frequently isolated, followed by LR genotypes 6 (18.9%) and 42 (17.8%). Table 3 displays results for the other variables. 

Baseline cytology results showed LSIL in 46.7%, normal mucosa in 40.8%, atypical squamous cells of uncertain significance (ASCUS) in 7.7%, and HISL in 2.6%. High-resolution anoscopy (HRA) results were normal in 52.3% of participants, LSIL (AIN1) in 38.5%, HSIL (AIN2/3) in 11%, and ASCC in 0.6% (Table 4).

### 2.2. Diagnostic Value of Cytology and HPV PCR for Anal Dysplasia 

LR-HPV PCR achieved 79.6% sensitivity (S), 27.9% specificity (Sp), 12.7% positive predictive value (PPV), and 91.6% negative predictive value (NPV) for the diagnosis of HSIL-plus, while cytology demonstrated 72.7% S, 43.7% Sp, 14.2% PPV, and 82.4% NPV for this diagnosis. HSIL-plus could be ruled out in all patients with anal mucosa monoinfection by LR-HPV genotypes and normal cytology (100% S, 91.9% Sp, 2.9% PPV, and 100% NPV) (Table 5). Correlations between biopsy, cytology, and HR and LR HPV PCR results for HSIL-plus never exceeded 0.4, the minimum value for analyzing agreement between techniques (Table 6).

### 2.3. Factors Related to the Progression of Lesions from LSIL to HSIL-Plus

The median follow-up period was 43 months (IQR: 12–76). After a mean follow-up of 12 months, progression from LSIL (AIN1) to HSIL-plus was observed in 23 participants (4.7%) (IQR: 12–12), with 1 progressing to invasive anal cancer (ASCC) and 22 to HSIL (AIN2/AIN3), while spontaneous regression of LSIL (AIN1) was recorded in 150 (30.4%) (IQR: 12–24). 

In bivariate analyses, the risk factors for progression from LSIL to HSIL-plus were age at HIV diagnosis (25.5 [IQR: 20.3–30.4] vs. 31 [IQR: 25.3–37] years; *p* = 0.016]), AIDS stage (43.5 vs. 23.8%; *p* = 0.036), non-retired (69.6 vs. 89.6%; *p* = 0.011), the presence of condylomas (43.5 vs. 20.3%; *p* = 0.009), a history of condylomas (56.5 vs. 29.3%; *p* = 0.007), the acquisition of HR (82.6 vs. 45.3%; *p* = 0.001) and LR (78.3 vs. 40.3%; *p* = 0.0001) genotypes, and anal mucosa infection by genotypes 6 (39.1 vs. 17%; *p* = 0.021) and 84 (21.7 vs. 7.1%; *p* = 0.029). Infection by HPV-62 genotype emerged as a protective factor against progression (0 vs. 16.4%, *p* = 0.033) (Table 7). 

In the multivariate analysis, progression from LSIL to HSIL-plus was related to the acquisition of HR (HR: 4.15; 95% CI 1.14–15.03) and LR (HR 3.68 95% CI: 1.04–12.94) HPV genotypes, including infection by genotype 6 (HR 4.47 95% CI: 1.34–14.91) and a history of AIDS (HR: 5.81 95% CI: 1.78–18.92) (Table 7).

In the study of factors favoring progression from AIN1 to HSIL-plus in patients with AIDS, no statistically significant association was found in bivariate analyses, which considered age, age at start of sexual relations, number of sexual partners in previous 12 months, VPH vaccination, time since HIV diagnosis, CD4 count, CD8 count, viral load, and number of HR and LR HPV genotypes and of all genotypes.

## 3. Discussion

The participants in this study were followed for a median of 43 months. One-third were aged under 30 years, and three-fifths between 30 and 50 years; around one-half of them smoked, and one-fifth reported a history of >300 sexual partners throughout life. In regard to HIV infection, their viroimmunological status was excellent, with only 4.1% in virologic failure and a median CD4 count of 698.9 cells; 71.2% were infected with LR genotypes, 74% with HR genotypes, and 56% were coinfected with LR and HR genotypes. Less than half of the participants had normal anal mucosa, around one-tenth had HSIL at enrolment, and three (0.6%) had anal cancer, a similar profile to that of other European cohorts [10], and even of a North American study population that mainly differed in ethnic make-up, with a predominance of Afro-Americans [11]

With respect to the value of LR-HPV PCR as the sole screening test for HSIL or anal cancer, it achieved S and NPV values that were higher than obtained with anal cytology and very similar to those observed with HR-HPV PCR. HSIL-plus could be ruled out in all participants with monoinfection by LR-HPV genotypes and normal cytology. This is the first report on the value of LR-HPV PCR in the diagnosis of these lesions in MSM-LHIV, showing that the combination of LR-HPV PCR and anal cytology ruled out HSIL-plus with 100% S and 100% NPV. These findings are in line with previously published results supporting the combination of HR-HPV PCR with anal cytology to screen for anal cancer and precursor lesions in HIV-positive MSM [12,13]

LSIL progressed to HSIL-plus (1 ASCC and 22 AIN2/3) in less than 5% of participants during the follow-up period, largely within one year of follow-up, lower than the progression rates of 16.4% [14] and 36% [15] described in previous studies. However, in comparison to the present study, the mean age was higher and a larger percentage of patients had CD4 nadir < 200 cells/uL in the former study [14] and the cohort was more heterogeneous in the latter, including men and women with and without HIV infection [15], which may in part account for the differences in progression rates; in addition, more than half of participants were smokers or ex-smokers in both previous studies, whereas less than half of the present patients were smokers and only 9.1% were ex-smokers. The objective of the present study was to investigate the screening, early diagnosis, treatment, and prophylaxis of anal dysplasia, resulting in a highly treated cohort. Thus, 15% had been vaccinated against HPV, and those with HSIL or intra-anal or genital condylomas had been treated with imiquimod, achieving HR-HPV and LR-HPV clearance rates of 30.4% and 43.5%, respectively, which would have contributed to the lower percentage who progressed to high-grade lesions or cancer of the anal mucosa. Participants were not vaccinated in the other two studies [14,15], those receiving treatment for HSIL-plus were excluded from one of them [14], and neither study described the virus clearance rate. In the present investigation, progression from LSIL to HSIL-plus in anal mucosa was associated with the acquisition of HR- and LR-HPV genotypes (specifically HPV-6) and a history of AIDS, while no association was observed with coinfection by HR- and LR-HPV genotypes. 

The degree of immunosuppression, defined by current CD4 count and CD4 nadir < 200 cells/uL, has been associated with the risk of lesion progression [16,17]. A recent HPV genotyping study of anal warts/condylomas in MSM-LHIV found that >20% of these lesions were HSIL and infected by LR and HR genotypes; the authors concluded that oncogenic and non-oncogenic HPV genotypes were involved in the genesis of HSIL-plus [18]. HR genotypes, in particular HPV-16, have frequently been related to anal lesion progression [19]. A meta-analysis published in 2018 described HPV-16 as the carcinogenic genotype *par excellence* in anal mucosa and responsible for both precursor lesions and anal cancer [20]. HPV-6 has historically been classified as an LR genotype and associated with the development of benign verrucose lesions or LSIL; however, an association was recently found between monoinfection by HPV-6 and progression to HSIL [21]. Furthermore, chronic monoinfection by LR genotypes 6, 11, 42, 44, or 70 has also been reported as the cause of a small number of anal cancers [22].

The results of this study are limited to a specific population of HIV-positive MSM and cannot be extrapolated to other types of subjects. Study strengths include the prospective and longitudinal design, large sample size, and systematic long-term follow-up. Notably, this is the first published report on monoinfection by LR-HPV genotypes in the anal canal of MSM-LHIV. 

In conclusion, monoinfection due to LR-HPV genotypes is not associated with anal cancer or precursor lesions in patients with normal cytology. The progression rate from LSIL to HSIL-plus was <5% and was related to the acquisition of HR- and LR-HPV genotypes (specifically HPV-6) and a history of AIDS. These findings are directly relevant to the care of anal dysplasia in MSM-LHIV, because the presence of LR-HPV genotypes alone with normal cytology rules out precancerous lesions and anal cancer, allowing for a longer interval between anoscopy examinations. Furthermore, given that only a very small percentage of patients with low-grade AIN1 progress, and over a prolonged period, these can be monitored by HRA every year rather than every three or six months. 

## 4. Materials and Methods

This prospective, longitudinal, single-center study included consecutive MSM-LHIV who participated between May 2010 and December 2021 in a program for the screening, diagnosis, treatment, and follow-up of anal mucosa dysplastic lesions. All participants were attended at Virgen de las Nieves University Hospital (HUVN) of Granada, after referral from Infectious Disease Units of the HUVN of Granada and Hospital Complex of Jaen (Jaen, Spain). This study was approved by the ethical committee of the coordinating center. All patients had given consent to the use of their data for research purposes, and all data were gathered in accordance with Spanish personal data protection legislation (Organic Law 3/5 December 2018).

The inclusion criterion was to be MSM-LHIV aged over 17 years. Exclusion criteria were being a heterosexual man or woman living with HIV and having a history of anal canal neoplasm in the previous five years.

At the baseline visit (V0), patients were informed about the conditions and objectives of this investigation, and data were gathered on their age, history of perianal-genital condylomas, number of sexual partners throughout life, number of sexual partners over past 12 months, age at onset of sexual relations, utilization of condoms and their percentage utilization, smoking (yes/no and packs/year), alcohol consumption (yes/no and standard drink units, SDUs), injection drug use (IDU), ex-IDU, nationality, schooling, months with HIV diagnosis, CDC-classified HIV stage, months under antiretroviral treatment (ART) and lines used, virological failure (RNA ≥ 50 copies/mL in ≥2 determinations during previous 6 months), CD4 nadir, CD4 and CD8 lymphocyte counts, viral load at HIV diagnosis, and the presence of other infections, including chronic hepatopathy by hepatitis B virus (HBV) or hepatitis C virus (HCV), syphilis, other sexual transmitted infections (STIs), perianal-genital warts, and latent, treated, or active tuberculous infection. Information was also collected on CD4 and CD8 cell counts, CD4/CD8 ratio, and viral load at V0. 

At the same visit, cotton swabs impregnated with physiological saline were used to take two anal canal mucosal samples: one for HPV detection and genotyping by qualitative polymerase chain reaction (PCR) (Linear Array HPV Genotyping Test) with a “GeneAmp PCR System 9700” thermocycler (Applied Biosystems, Roche, Switzerland), and the other for thin-layer cytological study with a ThinPrep 2000 Processor (Hologic, Mississauga, Canada). Both samples were immersed in thin-layer liquid and sent to the hospital pathology laboratory for analysis. Genotypes 16, 18, 26, 31, 33, 35, 39, 45, 51–53, 56, 58, 59, 66, 68, 73, and 82 were considered high risk (HR-HPV), and genotypes 6, 11, 34, 40, 42–44, 54, 55, 57, 61, 70–72, 81, 83, 84, and 89 low risk (LR-HPV). Genotypes 39, 45, 59, and 68 were classified as subspecies of genotype HPV 18 and genotypes 31, 33, 35, 52, 58, and 67 as subspecies of HPV 16 [23].

After an interval of 4–12 weeks, participants underwent rectal inspection and digital rectal examination followed by high-resolution anoscopy (HRA) with a Carl Zeiss 150 fc ^©^ colposcope (Carl Zeiss, Oberkochen, Germany), introducing a transparent disposable anoscope through which 5 mL acetic acid was instilled and left for around 3 min, removing it for examination of the mucosa. Next, 5% Lugol’s iodine was instilled for 1 min and the anoscopic examination was repeated. Samples were taken from quadrants of apparently normal mucosa and from areas with Lugol-negative aceto-white lesions. Biopsies were performed with endoscopic retrograde cholangiopancreatography (ERCP). 

Patients with normal anoscopy or LSIL(AIN1) were followed up at one year with cytology, HPV PCR, and anoscopy studies. Patients with HSIL either underwent mucosectomy by electrical scalpel (offered from May 2010 onwards) in the Coloproctology Unit of the General Surgery Department or self-administered 5% imiquimod three times a week for 16 weeks (offered from 2014 onwards). Anoscopy was performed in these patients at the end of their treatment, scheduling a follow-up at one year if the outcome was normal/LSIL or retreatment if HSIL, and referring the patient to the hospital oncology department if anal cancer was detected. Information was gathered at the follow-up visit on the number of sexual partners in the previous 12 months, the emergence of STIs (conducting syphilis serology in all patients but ordering PCR and anal-urethral exudate culture solely in patients with symptoms or infected partners), genital/anal condylomas, ART experience (change, virological failure, adherence rate), CD4 and CD8 lymphocyte counts, CD4/CD8 ratio, and HIV viral load. 

In the cytology study, the Bethesda classification [24] was used to categorize lesions as atypical squamous cells (ASC), atypical squamous cells—high (ASC-H), LSIL, or HSIL. The study variable “·abnormal cytology” includes ASCUS, LSIL, or HSIL. In the histology study, the proposal of the Lower Anogenital Squamous Terminology (LAST) Standardization Project for HPV served to classify lesions as LSIL (AIN1/condyloma), HSIL (AIN2, AIN3, C. in situ), or invasive carcinoma (ASCC) [25]. The study variable “HSIL plus” includes anal lesions ranging from HSIL to invasive cancer (high-grade SIL/cancer).

### Statistical Analysis

In descriptive analysis, means, standard deviations, medians, and percentiles were calculated for quantitative variables and absolute and relative frequencies for qualitative variables. In bivariate analyses, the Student’s *t*-test for independent samples was applied for quantitative variables when normally distributed according to the Kolmogorov–Smirnov test and the Mann–Whitney U test when non-normally distributed. Qualitative variables were analyzed with Pearson’s chi-square test or, when application criteria were not met, Fisher’s test. Multiple logistic regression analysis was then performed, entering variables that were significant in bivariate analyses or considered relevant in the literature. SPSS 21.0 (IBM SPSS, Armonk, NY, USA) was used for data analyses, and the level of significance was 0.05 in all tests. 

## Figures and Tables

**Table 1 ijms-24-05642-t001:** Epidemiological characteristics of the cohort.

	Cohort N = 493
Age, mean (years), (±SD) <30 years Between 30 and 50 years >50 years	36.48 (±10.434) 151 (30.6) 290 (58.8) 52 (10.5)
Nationality, n (%) European Central/South American African	464 (94.1) 27 (5.5) 2 (0.4)
Median NPT (IQR) Median NP12m (IQR) Median age at start of sexual relations ((IQR) Use of condom, n (%) Median % of use of condom (IQR) Relationships during the previous year, n (%)	50 (20–200) 2 (1–8) 18 (16–20) 329 (66.7) 100 (0–100) 442 (89.7)
Occupation (%) Non-retired (including students, actively employed, unemployed, etc.) Schooling level, n (%) No schooling Primary Secondary University	434 (88) 5 (3.5) 24 (16.9) 40 (28.2) 69 (48.6)
Smoking, n (%) Ex-smoker, n (%) Median packs/year, (IQR) Alcohol, n (%) Median SDUs, (IQR) Ex-IDU, n (%) Polypharmacy, n (%)	237 (48.1) 45 (9.1) 1.65 (0–15) 210 (42.6) 0 (0–4) 4 (0.8) 55 (11.2)
HPV vaccination, n (%)	74 (15)
Chronic hepatopathy by HBV, n (%) Chronic hepatopathy by HCV, n (%) Syphilis in baseline visit, n (%) Other STIs at baseline visit, n (%) Total number of STIs at baseline visit, n (%) - None - One - Two - Three Positive Mantoux at baseline visit, n (%) Comorbidities, n (%)	16 (3.2) 14 (2.8) 154 (31.2) 144 (29.2) 288 (58.4) 196 (39.8) 6 (1.2) 2 (0.4) 33 (6.7) 60 (12.2)
Condylomas in inclusion visit, n (%) History of genital condylomas, n (%)	105 (21.3) 149 (30.2)

NP12m, number of sexual partners during the previous 12 months; NPT, total number of sexual partners since initiation of sexual relations; SRs, sexual relationship; SDUs, standard drink units; IDU, injection drug user; Ex-IDU, ex-infection drug user; HPV, human papillomavirus; HBV, hepatitis B virus; HCV, hepatitis C virus; TBC, tuberculosis; STI, sexually transmitted infection.

**Table 2 ijms-24-05642-t002:** Variables related to HIV infection.

	Cohort N = 493
Risk for acquiring HIV infection - MSM - IDU - Unknown	485 (98) 1 (0.2) 6 (1.2)
Median months since diagnosis of HIV infection (IQR) Median age at diagnosis of HIV infection (IQR) AIDS (A3, B3, C), n (%)	30.5 (10–105) 31 (25–37) 114 (23.1)
CD4 at diagnosis, mean (±SD) CD8 at diagnosis, mean (±SD) VL at diagnosis, (log10) mean (±SD) Nadir, n (%) <200 200–500 >500 CD4 baseline visit, mean (±SD) CD8 baseline visit, mean (±SD) CD4/CD8 baseline visit, median (IQR) VL (log10) baseline visit, mean (±SD) VL < 50 cop/mL baseline visit, n (%)	444.58 (±288.8) 1115.57 (±749.9) 5.5 (±6.1) 114 (231) 231 (46.9%) 129 (26.2) 689.8 (±450.5) 1954.4 (±528.7) 0.7(0.46–0.98) 3.6 (±4.2) 344 (70.9)
Naïve, n (%) Median ART lines from initiation (IQR) Median months with ART up to V0 (IQR) ART line baseline visit, median, (IQR) Adherence to ART at V0, median % adherence (IQR) Virological failure, n (%) Reason for ART abandonment, n (%) - Adverse effects - Other causes ART, n (%) - NRTI - NNRTI - PI - INSTI	53 (10.8) 1 (1–2) 20 (7–73) 1 (1–2) 100 (100–100) 20 (4.1) 8 (1.6) 5 (1) 393 (79.7) 162 (32.9) 129 (26.2) 190 (38.5)

HIV, human immunodeficiency virus; AIDS, acquired immunodeficiency syndrome; MSM, men who have sex with men; IDU: injection drug user; CD4, CD4 lymphocytes; CD8, CD8 lymphocytes; VL, viral load; CD4/CD8, CD4/CD8 ratio; ART, antiretroviral therapy; NRTI, nucleoside analog reverse transcriptase inhibitor; NNRTI, non-nucleoside analog reverse transcriptase inhibitor; PI, protease inhibitor; INSTI, integrase strand transfer inhibitors.

**Table 3 ijms-24-05642-t003:** HPV infection data at baseline visit.

	N = 493
HR-HPV, n (%) Number of HR-HPV serotypes, median (P25–P75) LR-HPV, n (%) Number of LR-HPV serotypes, median (P25–P75) HPV coinfection (HR and LR), n (%) A9 clade: HPV 18, 39, 45, 59, 60, n (%) A7 clade: HPV 16, 31, 33, 35,52, 58, 67, n (%) Simultaneous infection oncogenic HPV clades A7 and A9, n (%) HPV 6, n (%) HPV 11, n (%) HPV 16, n (%) HPV 18, n (%) HPV 26, n (%) HPV 31, n (%) HPV 33, n (%) HPV 35, n (%) HPV 39, n (%) HPV 40, n (%) HPV 42, n (%) HPV 43, n (%) HPV 45, n (%) HPV 51, n (%) HPV 52, n (%) HPV 53, n (%) HPV 54, n (%) HPV 55, n (%) HPV 56, n (%) HPV 58, n (%) HPV 59, n (%) HPV 61, n (%) HPV 62, n (%) HPV 66, n (%) HPV 68, n (%) HPV 69, n (%) HPV 70, n (%) HPV 72, n (%) HPV 73, n (%) HPV 81, n (%) HPV 82, n (%) HPV 83, n (%) HPV 84, n (%) HPV 108, n (%)	365 (74) 1 (1–3) 351 (71.2) 1 (0–2) 276 (56) 206 (41.8) 238 (48.3) 119 (24.1) 93 (18.9) 82 (16.6) 130 (26.4) 64 (13) 8 (1.6) 67 (13.6) 36 (7.3) 48 (9.7) 61 (12.4) 12 (2.4) 88 (17.8) 20 (4.1) 63 (12.8) 71 (14.4) 63 (12.8) 42 (8.5) 36 (7.3) 83 (16.8) 40 (8.1) 35 (7.1) 50 (10.1) 35 (7.1) 84 (17) 40 (8.1) 55 (11.2) 16 (3.2) 38 (7.7) 31 (6.3) 44 (8.9) 79 (16) 25 (5.1) 5 (1.0) 33 (6.7) 13 (2.6)

**Table 4 ijms-24-05642-t004:** Cytology and anoscopy results at baseline visit.

Cytology	N = 493
Normal, n (%) - Normal, without HR genotypes, n (%) - Normal, with HR genotypes, n (%) ASCUS, n (%) LSIL, n (%) HSIL, n (%)	201 (40.8) 42 (8.5) 149 (30.2) 38 (7.7) 230 (46.7) 13 (2.6)
High-resolution anoscopy (HRA)	N = 493
Abnormal, n (%) LSIL (AIN1), n (%) HSIL (AIN2/3), n (%) Anal carcinoma (ASCC)	235 (47.7) 190 (38.5) 54 (11) 3 (0.6)

LSIL, low-degree squamous intraepithelial lesion; HSIL, high-degree squamous intraepithelial lesion; ASCUS, atypical squamous cells of undetermined significance; HRA high-resolution anoscopy; HPV, human papillomavirus; HR, high-risk genotype; LR, low-risk genotype.

**Table 5 ijms-24-05642-t005:** Sensitivity, specificity, PPV, and NPV of anal cytology and HPV PCR for diagnosis of HSIL-plus.

	Sensitivity (%)	Specificity (%)	PPV (%)	NPV (%)
Normal Cytology (n = 202)	27.3	56.3	7.4	85.7
Abnormal Cytology (n = 281)	72.7	43.7	14.2	82.4
LSIL (n = 230)	61.8	54.2	14.8	91.7
HSIL (n = 13)	12.7	98.6	53.8	89.7
ASCUS (n = 38)	1.8	91.3	2.6	87.8
HPV—High-risk (n = 368)	85.7	25.4	13	93.2
HPV—Low-risk (n = 43)	79.6	27.9	12.7	91.6
HPV—High- and low-risk (n = 277)	66.1	43.9	13.4	90.8
HPV—Low-risk positive and normal cytology (n = 125)	14.8	72.3	6.4	86.9
HPV—Low-risk positive, HPV—High-risk negative, and normal cytology (n = 35)	100	91.9	2.9	100
HPV—Low-risk negative and normal cytology (n = 75)	9.4	83.4	6.7	88
HPV—High-risk negative and normal cytology (n = 61)	100	90.2	87.5	100

LSIL: low-grade intraepithelial lesions; HSIL: high-grade intraepithelial lesions; ASCUS: atypical squamous cells of unknown significance.

**Table 6 ijms-24-05642-t006:** Correlation of anal HPV cytology and PCR with histology.

	Normal		LSIL (AIN 1)		HSIL (AIN2 and 3)		SCCA	
	N = 250		N = 189		N = 54		N = 1	
	n (%)	*p* *	n (%)	*p* *	n (%)	*p* *	n (%)	*p* *
	Kappa	Spearman	Kappa	Spearman	Kappa	Spearman	Kappa	Spearman
	136 (54.4)	0.0001	55 (29.1)	0.0001	13 (24.1)	0.005	1 (100)	0.3
Normal Cytology (n = 201)	0.26	0.26	0.21	0.21	0.09	0.13	0.006	0.054
	101 (40.4)	00001	134 (70.9)	0.0001	41 (75.9)	0.005	0	0.3
Abnormal Cytology (n = 281)	0.29	0.3	0.21	0.21	0.09	0.13	0.006	0.054
	82 (32.8)	0.0001	118 (62.4)	0.0001	35 (64.8)	0.007	0	0.34
LSIL (n = 230)	0.31	0.31	0.23	0.24	0.08	0.12	0.004	0.04
	3 (1.2)	0.036	3 (1.6)	0.23	7 (13)	0.001	0	1
HSIL (n = 13)	0.032	0.095	0.02	0.05	0.17	0.23	0.004	0.008
	24 (9.6)	0.143	16 (8.5)	0.69	1 (1.9)	0.8	0	1
ASCUS (n = 38)	0.037	0.067	0.011	0.02	0.08	0.08	0.004	0.01
	170 (68)	0.0001	161 (85.2)	0.0001	47 (87)	0.04	2 (100)	1
HPV—High-risk (n = 368)	0.16	0.19	0.13	0.17	0.04	0.09	0.003	0.04
	170 (68)	0.012	147 (77.8)	0.055	43 (79.6)	0.24	2 (100)	1
HPV—Low-risk (n = 353)	0.1	0.12	0.07	0.09	0.02	0.05	0.003	0.04
	122 (49)	0.0001	126 (66.7)	0.001	36 (66.7)	0.14	2 (100)	0.5
HPV—High- and low-risk (n = 277)	0.17	0.17	0.14	0.15	0.04	0.07	0.006	0.06
	49 (19.6)	0.0001	11 (5.9)	0.001	0	0.003	0	0.7
HPV—High-risk negative and normal cytology (n = 61)	0.15	0.22	0.13	0.17	0.14	0.14	0.004	0.02
	81 (33.2)	0.0001	39 (20.7)	0.027	8 (14.8)	0.04	1 (100)	0.09
HPV—Low-risk positive and normal cytology (n = 125)	0.14	0.162	0.09	0.1	0.08	0.09	0.012	0.08
	28 (11.5)	0.0001	6 (3.2)	0.007	0	0.023	0	0.8
HPV—Low-risk positive, HPV—High-risk negative, and normal cytology (n = 34)	0.09	0.17	0.076	0.124	0.096	0.09	0.004	0.013
	51 (20.9)	0.002	19 (0.1)	0.006	5 (9.4)	0.18	0	0.7
HPV—Low-risk negative and normal cytology (n = 75)	0.11	0.14	0.11	0.13	0.06	0.06	0.004	0.02

ASCUS: atypical squamous cells of unknown significance; LSIL: low-grade intraepithelial lesions; HSIL: high-grade intraepithelial lesions; AIN: anal intraepithelial neoplasia. SCCA: squamous cell carcinoma of the anus. *p* * > 0.005; correlation (kappa and Spearman).

**Table 7 ijms-24-05642-t007:** Results of bivariate and multivariate analyses.

	Progressor N = 23	Non-Progressor N = 317	Bivariate *p**	Multivariate HR, 95% IC
Mean age (years), (±DS) Retired, n (%) Schooling level - No schooling - Primary - Secondary—Vocational - University Nationality, n (%) - European, n (%) - Central/South American, n (%) - African, n (%) Active smoking Alcohol	32.43 (8.98) 2 (8.7) 0 (0) 4 (17.4) 7 (30.4) 12 (52.2) 21 (91.3) 2 (8.7) 0 (0) 13 (56.5) 10 (43.5)	36.79 (10.36) 17 (5.4) 6 (1.9) 36 (11.4) 95 (30.1) 179 (56.6) 300 (94.6) 16 (5) 1 (0.3) 144 (45.4) 139 (43.8)	0.051 0.374 0.761 0.727 0.303 0.963	0.97 (0.87–1.06)
Use of condom, n (%) Sexual relations during previous year, n (%)	2 (8.7) 22 (95.7)	91 (29.4) 282 (89.2)	0.096 0.490	2.81 (0.58–13.57)
Previous history of - Hepatopathy by HCV - Hepatopathy by HBV - Syphilis, n (%) - Other STIs, n (%) - Montoux-positive, n (%) History of condylomas, n (%) Current condylomas, n (%)	2 (8.7) 0 (0) 9 (39.1) 8 (34.8) 2 (8.7) 13 (56.5) 10 (43.5)	6 (1.9) 12 (3.8) 87 (27.6) 87 (27.4) 24 (7.7) 92 (29.3) 64 (20.3)	0.096 1 0.237 0.449 0.696 0.007 0.009	1.29 (0.39–4.21) 1.74 (0.51–5.96)
CD4 nadir (cells/uL), (±SD) AIDS stage (A3, B3, C), n (%) CD4 at diagnosis, mean (±SD) CD8 at diagnosis, mean (±SD) VL at diagnosis (log), mean (±SD) Baseline CD4/CD8, mean (±SD) Baseline CD4, mean (±SD) Baseline CD8, mean (±SD) Baseline VL of HIV (log), mean (±SD)	380.6 (211.9) 10 (43.5) 474.5 (285.8) 1055.6 (737.4) 4.7 (4.84) 0.65 (0.24) 642.6 (309.7) 1022.6 (369) 3.78 (3.3)	368.2 (233.8) 75 (23.8) 439.8 (293.8) 1161.7 (831.4) 5.5 (6.11) 2.45 (11.8) 704 (516.4) 1083.7 (545.9) 3.5 (4.2)	0.806 0.036 0.593 0.638 0.349 0.465 0.574 0.598 0.454	5.81 (1.78–18.92)
ART, n (%) Virological failure, n (%)	21 (91.3) 3 (15)	285 (89.9) 14 (4.9)	1 0.91	
Median age at HIV diagnosis, years (IQR) Median time since HIV diagnosis, months (IQR)	25.5 (20.3–30.4) 28.5 (14.5–69)	31 (25.3–37) 28 (8–98.75)	0.016 0.5	0.97 (0.86–−1.10)
Median number of LR-HPV Median number of HR-HPV LR-HPV, n (%) HR-HPV, n (%) Coinfection HR and LR-HPV, n (%) Oncogenic HPV clades (A7 plus A9) HPV 6 HPV 11 HPV 16 HPV 18 HPV 31 HPV 33 HPV 35 HPV 39 HPV 42 HPV 45 HPV 51 HPV 52 HPV 55 HPV 56 HPV 58 HPV 59 HPV 61 HPV 62 HPV 66 HPV 68 HPV 70 HPV 72 HPV 81 HPV 82 Clearance HR-HPV, n (%) Clearance LR-HPV, n (%) Acquisition HR-HPV, n (%) Acquisition LR-HPV, n (%)	1 (0.75–3) 1 (1–3) 18 (78.3) 19 (82.6) 15 (65.2) 5 (21.7) 9 (39.1) 3 (13) 4 (17.4) 4 (17.4) 2 (8.7) 2 (8.7) 1 (4.3) 1 (4.3) 5 (21.7) 4 (17.4) 5 (21.7) 1 (4.3) 2 (8.7) 3 (13) 0 (0) 3 (13) 1 (4.3) 0 (0) 4 (17.4) 2 (8.7) 2 (8.7) 3 (13) 3 (13) 5 (21.7) 7 (30.4) 10 (43.5) 19 (82.6) 18 (78.3)	1 (1–2) 2 (1–3) 222 (71.2) 239 (76.6) 177 (56.9) 76 (24.4) 53 (17) 57 (18.3) 80 (25.8) 36 (11.6) 51 (16.4) 28 (9) 33 (10.6) 37 (11.9) 54 (17.4) 40 (12.9) 41 (13.2) 42 (13.5) 57 (18.3) 29 (9.3) 21 (6.8) 31 (10) 23 (7.4) 51 (16.4) 30 (9.6) 27 (8.7) 24 (7.7) 21 (6.8) 45 (14.5) 22 (7.1) 101 (33.9) 84 (27.7) 140 (45.3) 124 (40.3)	0.42 0.61 0.46 0.50 0.43 0.77 0.021 0.778 0.37 0.50 0.552 1 0.49 0.49 0.57 0.52 0.341 0.333 0.394 0.473 0.38 0.717 1 0.033 0.273 1 0.697 0.223 1 0.029 0.735 0.108 0.001 0.0001	4.47 (1.34–14.91) 0 (0–0) 0.43 (0.104–1.76) 4.15 (1.14–15.03) 3.68 (1.04–12.94)

HBV, hepatitis B virus; HCV, hepatitis C virus; STI, sexually transmitted infection; HRA, high-resolution anoscopy, HIV, human immunodeficiency virus; AIDS, acquired immunodeficiency syndrome; CD4, CD4 lymphocytes; CD8, CD8 lymphocytes; VL, viral load; CD4/CD8, CD4/CD8 ratio; ART, antiretroviral therapy; HPV, human papillomavirus; HR, high-risk genotype; LR, low-risk genotype.

## Data Availability

The researchers confirm the accuracy of the data provided for the study, as well as their availability.

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
