# Peer review of "Role of Low-Risk HPV PCR Monoinfection in Screening for HSIL and Anal Cancer in Men Who Have Sex with Men Living with HIV"

_ijms, 2023, doi:10.3390/ijms24065642_

Round 1
Reviewer 1 Report
This study investigate the role of low risk HPV monoinfection in HSIL and anal cancer in men who have sex with men living with HIV in Spain. Interestingly the findings indicate that progression from LSIL to HSIL-plus was associated with genotype acquisition of HR as well as LR HPV genotypes, especially 6, and advancing history of AIDS.
Whether the co-infection of both HR and LR HPVs was associated with the progression to anal cancer is not clearly addressed?
There was no comparison of data solely done with AIDS patients to exclude the influence of HIV/AIDS to the speed of cancer progression.
The above limitations should be addressed in the Discussion.
Minor Concern
Some Tables are very lengthy with too much information.
The order of criteria and numbers/values is not aligned in my copy. So it was hard to delineate results.
Author Response
RESPONSES TO REVIEWER 1
This study investigated the role of low risk HPV monoinfection in HSIL and anal cancer in men who have sex with men living with HIV in Spain. Interestingly the findings indicate that progression from LSIL to HSIL-plus was associated with genotype acquisition of HR as well as LR HPV genotypes, especially 6, and advancing history of AIDS.
1- Whether the co-infection of both HR and LR HPVs was associated with the progression to anal cancer is not clearly addressed?
The manuscript has been modified accordingly, providing further detail on the 23 progressions (1 to invasive anal cancer and 22 to HSIL (AIN2(Ain3). It is now stated that no significant association was found with coinfection (p=0,43); coinfection was previously reported as “mixed infection” in table 6; this term has now been changed.
2- There was no comparison of data solely done with AIDS patients to exclude the influence of HIV/AIDS to the speed of cancer progression.
This was not included because none of the study variables showed a statistically significant association with progression of AIN1 to HSIL-plus in these patients, as now reported in the revised paper, Lines 167-171
3- The above limitations should be addressed in the Discussion.
The Discussion has been expanded to address these points (lines 194-213)
Minor Concern
4-Some Tables are very lengthy with too much information.
We have reduced the length of the tables, removing non-essential information.
5- The order of criteria and numbers/values is not aligned in my copy. So it was hard to delineate results.
We apologize for this problem, which has now been corrected.

Reviewer 2 Report
This study is a statistical analysis, there are many tables but lack of brief description of results. The Discussion is not intensive and not comprehensive. Reference format is not consistent.
Author Response
RESPONSES TO REVIEWER 2
1- This study is a statistical analysis, there are many tables but lack of brief description of results.
We now provide more information on key results in the revised text. (lines 66-70, 78-81, 112-113, 139-140, 167-171)
2- The Discussion is not intensive and not comprehensive.
We have revised and expanded the Discussion accordingly. In fact, there have been few relevant studies, and we comment on all of them, contrasting our findings with those of comparable studies. The Discussion has been expanded to address these points (lines 194-213)
3- Reference format is not consistent.
References now consistently follow Vancouver guidelines.
- English style
The paper has been re-revised by Richard Davies, graduate of Cambridge University with more than 30 years’ experience successfully preparing papers for high-impact medical journals. A certificate is attached confirming the quality of the writing (US English style).

Round 2
Reviewer 2 Report
The manuscript has been improved although not satisfied. Authors may also address the implications or suggestions from these results in the Discussion.
I would like to recommend the tables can be reformatted for friendly reading form.
Author Response
Dear Editor:
We have addressed the concerns raised by Reviewer 2 (see below) and trust that our paper can now be accepted for publication.
RESPONSE TO REVIEWER 1
We are grateful for the positive appreciation of Reviewer 1.
RESPONSE TO REVIEWER 2
Reviewer 2:
Comment: The manuscript has been improved although not satisfied.
Authors may also address the implications or suggestions from these results in the Discussion.
Response: We have expanded our Conclusions to highlight the implications and clinical relevance of our findings, as requested (lines 217-222)
Comment: I would like to recommend the tables can be reformatted for friendly reading form.
Response: We have followed this recommendation, reformatting (dividing) the tables to facilitate their reading.
